pedology/environmental science/geophysics

natural capital, soil function, soil properties, valuation

**Author for correspondence:**
Aline F. Rodrigues
e-mail: frodriguesaline@gmail.com

# Systematic review of soil ecosystem services in tropical regions

Aline F. Rodrigues[1,2], Agnieszka E. Latawiec[1,2,3,4], Brian J. Reid[4], Alexandro Solórzano[1], Azeneth E. Schuler[5], Carine Lacerda[1], Elaine C. C. Fidalgo[5], Fabio R. Scarano[6,7], Fernanda Tubenchlak[2], Ingrid Pena[1,2], Jose Luis Vicente-Vicente[8], Katarzyna A. Korys[2], Miguel Cooper[9], Nelson F. Fernandes[10], Rachel B. Prado[5], Veronica Maioli[2], Viviane Dib[2,6] and Wenceslau G. Teixeira[5]

[1]Department of Geography and Environment – Rio Conservation and Sustainability Science Centre, Pontifical Catholic University of Rio de Janeiro, R. Marquês de São Vicente, 225 – Gávea, Rio de Janeiro, RJ 22451-000, Brazil
[2]International Institute for Sustainability, R. Dona Castorina 124 22460-320, Rio de Janeiro, Brazil
[3]Department of Production Engineering, Logistic and Applied Computer Sciences, Faculty of Production and Power Engineering, University of Agriculture in Kraków, Balicka 116B, 30-149, Kraków, Poland
[4]University of East Anglia, Norwich Research Park, Norwich NR4 7TJ, UK
[5]Embrapa Soils, R. Jardim Botânico, 1024, Rio de Janeiro, RJ 22460-000, Brazil
[6]Department of Ecology, Federal University of Rio de Janeiro, Rio de Janeiro, RJ, Brazil
[7]Brazilian Platform on Biodiversity and Ecosystem Services – BPBES, Campinas, SP, Brazil
[8]Leibniz Centre for Agricultural Landscape Research, Eberswalder Str. 84 15374 Müncheberg, Germany
[9]Department of Soil Science, University of São Paulo/ESALQ, Pádua Dias Av. 1, Piracicaba, SP 13418-900, Brazil
[10]Department of Geography, Federal University of Rio de Janeiro, Rio de Janeiro, RJ, Brazil

AFR, 0000-0003-3231-551X; AEL, 0000-0003-3036-1870;
AS, 0000-0001-7562-0720; AES, 0000-0002-8855-4352;
CL, 0000-0003-2253-5316; ECCF, 0000-0003-3648-1662;
FRS, 0000-0003-3355-9882; FT, 0000-0002-0223-3559;
IP, 0000-0002-4717-9234; JLV-V, 0000-0003-3554-9354;
KAK, 0000-0002-4301-2146; MC, 0000-0003-4922-4657;
NFF, 0000-0003-4747-3342; RBP, 0000-0002-1893-4915;
VM, 0000-0003-0839-8765; VD, 0000-0001-5457-5983;
WGT, 0000-0001-5493-1909

Soil ecosystem service (SES) approaches evidence the importance of soil for human well-being, contribute to improving dialogue between science and decision-making and encourage the translation of scientific results into public policies. Herein, through systematic review, we assess the state of the art of SES approaches in tropical regions. Through this review, 41 publications were identified; while most of these studies considered SES, a lack of a consistent framework to define SES was apparent. Most studies measured soil natural capital and processes, while only three studies undertook monetary valuation. Although the number of publications increased (from 1 to 41), between 2001 and 2019, the total number of publications for tropical regions is still small. Countries with the largest number of publications were Brazil ($n = 8$), Colombia ($n = 6$) and Mexico ($n = 4$). This observation emphasizes an important knowledge gap pertaining to SES approaches and their link to tropical regions. With global momentum behind SES approaches, there is an opportunity to integrate SES approaches into policy and practice in tropical regions. The use of SES evaluation tools in tropical regions could transform how land use decisions are informed, mitigating soil degradation and protecting the ecosystems that soil underpins.

# 1. Introduction

Soils are a limited natural resource, which support multiple ecosystems functions and associated services [1]. Relationships, at different spatial and temporal scales, across human and non-human interfaces sustain a cultural connection between soils, ecosystems and life [2]. Soils support human life through the production of food, wood, fibre and purification of water, and support the delivery of other diverse ecosystem services (ES) [3–5]. In acknowledgement of land degradation, and the importance of soils in sustaining ecosystems and human life, the services provided by soils have been the focus of recent discussions in the scientific literature [3,6–8].

ES approaches emerged at the end of the last century to encourage better dialogue between science and decision-making, to provide further prominence between ecosystems and human needs and to help translate scientific results into a format relevant to environmental governance and public policies [9,10]. Several conceptual frameworks aiming at a better understanding of ES have been developed [11–14]. These concepts have gained momentum in the literature in recent years and have also generated criticism regarding their anthropocentric framing [15]. One of the outcomes of this challenge has been the development of the concept of nature's contribution to people (NCP) as a more comprehensive definition. NCP considers that culture plays a vital role in the links between people and non-human nature and the sense of understanding that non-human nature can provide benefits and harms to human beings [14,16]. In the frameworks that address both concepts, soils have been little discussed and largely overlooked [6,8,17].

A systematic review highlighted that until 2008, the concept of ES related to soil quality received less attention when compared to biodiversity [18]. In the last 10 years, several authors have proposed and developed conceptual frameworks and classifications of soil ecosystem services (SES) (figure 1) [3,6,7,9]. In these pioneering works, the importance of defining the concept of SES is highlighted. A systemic issue in reaching a consensus on SES definition is that soil services have been used as a synonym for soil functions and soil processes [7,20]. This imprecision in framing has blurred the lines. Semantically, the word 'service' is defined as the supply of something that people need [21]. On the other hand, 'function' is the purpose of something or someone, and 'processes' a series of connected changes [21,22]. SES can be understood as flows of soil natural capital stocks that benefit humans and can be classified into regulation, provision and cultural [6]. Soil functions can be defined as flows arising from natural capital stocks that benefit all nature (human and non-human) [19]. Also, services, functions and processes are bounded by the properties of soil. In general, properties are directly measurable and express chemical (for example, pH), physical (density and aggregation) and biological (floral and faunal communities) characteristics. Soil processes are understood as the transformation of inputs into products, for example, the decomposition of organic matter to form humus, the compaction of the soil that reduces infiltration and promotes water run-off [6]. Soil service, function, processes and properties are summarized in figure 1.

The high demand for services provided by soils has transformed the global landscape [1]. Natural ecosystems have been degraded and/or replaced by agricultural production, mining activities and cities [23]. The impacts of such changes are projected to worsen with the increasing global population, which, according to United Nations projections, may reach 9.7 billion inhabitants in 2050 [24]. Sustainable soil management along with conservation and restoration initiatives, alongside changing

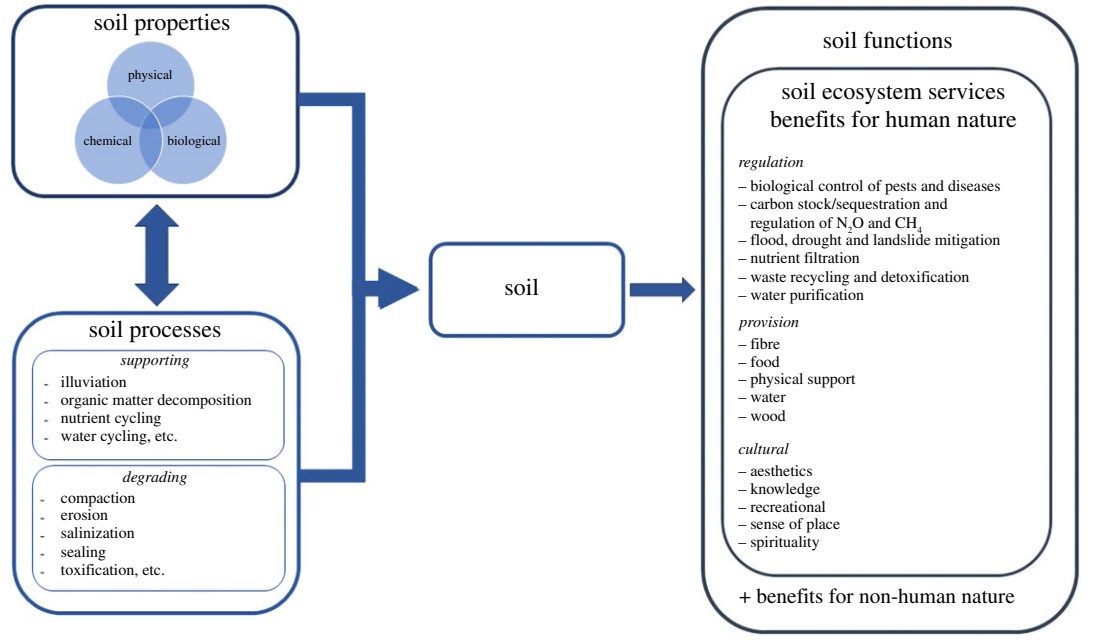

**Figure 1.** Illustrative framework of SES and their associated concepts—soil properties, soil process and soil functions. This framework is based on ideas contained within Dominati *et al.* [6]; Robinson *et al.* [9]; Baveye *et al.* [3] and Baveye *et al.* [19].

society's consumption habits, are urgently needed to assure the provision of SES. The absence of effective measures to prevent further soil degradation will lead to intensification of climate change, hunger and the spread of diseases [3,5,6].

Important tropical forests, such as the Amazon, are threatened by deforestation and fragmentation. These forests usually house traditional populations deeply connected and dependent on forest resources [25,26]. In addition, the tropical region houses countries (such as Brazil, India and Indonesia) notable for their production of agricultural products [27]. The advance of the agricultural frontier is often accompanied by soil degradation and its associated environmental and economic impacts [28,29]. In addition, changes in land use and cover and heavy rainfall in the tropical region have contributed significantly to soil erosion [30]. Improving this socio-ecological scenario is a huge social and political challenge, as countries in the tropical region need to accommodate population growth, food security, economic development, ecosystem conservation and the provision of ES [31]. The SES approach has the potential to contribute to divert this scenario of soil degradation.

Through a systematic literature review, we assessed the state of the art of SES in tropical regions. In order to guide future research and to encourage a more robust critical view on evaluating and valuing soil services, this study identifies knowledge gaps and problems associated with the use of the SES concept in the tropical region. We investigated (i) how the concept was used in the studies that align with SES concept, (ii) how SES have been evaluated and valued, and (iii) trends in publication over time and geographic distribution of the studies.

## 2. Material and methods

The systematic literature review was carried out through Web of Science, Scopus and Scielo database searching for articles published until April 2019. The following keyword combinations were used in the advance search engine: TS (topic) = (ecosystem service AND soil AND tropical region). We recognize that the term 'tropical region' may have limited the search, as many articles use other spatial cut-outs (example country names) to define their locations. However, there are more than 90 countries in the tropical region and considering all of them as search terms would render this systematic review unfeasible from a practical standpoint. All articles found were read ($n = 161$). The criteria for articles inclusion in the systematic review database were (i) articles consider tropical regions (23.27° North and 23.27° South); (ii) articles must mention the term 'Ecosystem Services'; (iii) articles must be concerned with soil; and (iv) the terms ES and soil must be related (see electronic supplementary material). The articles that met the criteria were exported to Mendeley ($n = 41$) for the extraction of information (electronic supplementary material) (figure 2). We recognize that soil ES

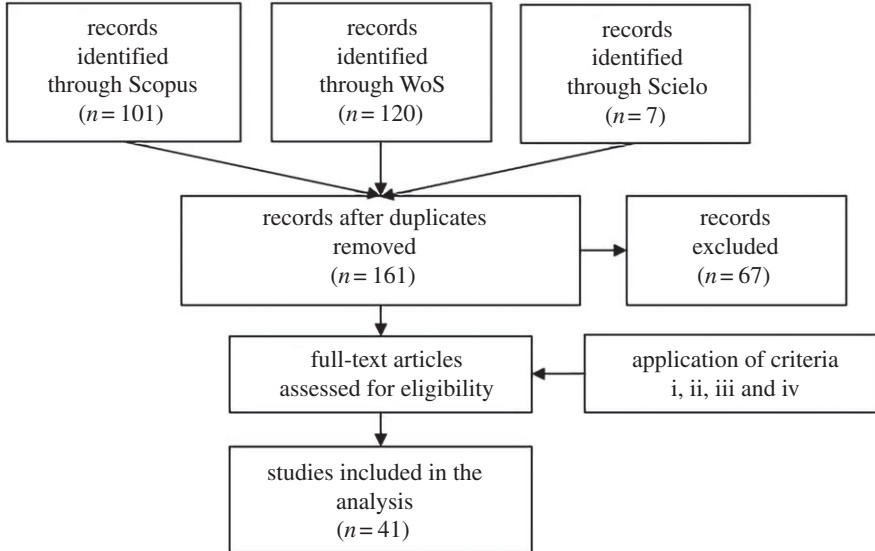

**Figure 2.** Systematic review flow diagram.

discussion may have been explored in other articles and in the grey literature, that are absent in the database assessed in this research (we acknowledge this represents a methodological limitation). Also, there are likely to be published studies that addressed SES that do not present the terms 'ecosystem services' linked with 'soil'. These publications were not included in our database because they did not use the terms that are part of the main research topic. The following information was extracted from the downloaded articles: year of publication, country, location of the studies (coordinates), if the SES were defined as flows, which SES were addressed and ES classification (regulation, provision and cultural—figure 1), methods of assessing the SES and valuation (see electronic supplementary material). All graphs were made using the SigmaPlot software v. 14 Trial Version. The map was made in QGIS 3.4.4 to spatialize the number of studies by countries. The word cloud of the SES most cited in the studies was made using the WordArt software.

# 3. Results and discussion

## 3.1. Classification of soil ecosystem services in the case studies

Of the 41 articles evaluated, 36 provided some classification of SES. Among them, 17% ($n = 7$) of the studies classified SES as services (understood here as the flow of soil natural capital stocks beneficial to people); 41% ($n = 17$) of the studies classified SES as soil services, processes, properties and functions; 29% ($n = 12$) classified SES as soil processes, properties and functions and 12% ($n = 5$) refer to articles that related the SES concept to aspects of the soil that do not fit into any of the categories (figure 3). Since its emergence, the ES concept has faced challenges for an accurate definition [11,13,14,32]. The Economics of Ecosystems and Biodiversity (TEEB), for example, removed support services from its framework as they do not directly benefit society and defined them as biophysical structure, processes and functions [33]. As noted previously, in many studies, the term SES has been used interchangeably with the terms soil function and soil processes [4,6,34]. This observation highlights the need for adopting a clear and universal definition of SES.

The most addressed SES in the studies were carbon stock ($n = 8$), food provision ($n = 6$), carbon sequestration ($n = 4$), flood mitigation ($n = 4$), water purification ($n = 4$) and water provision ($n = 4$)—classified as regulation and provision SES (figure 4). Previous studies demonstrated that landscape transformation and climate change could affect the tropical region more intensely, leading to loss of biodiversity, causing more intense rain in certain regions and affecting food production [35–37]. This fact may explain the greater attention of studies to these SES. Also, the market interest in the supply of food, carbon stock and sequestration may have contributed to greater interest in these services. In a global scale review of SES, the authors also found that most of the services covered were those of regulation and provision [8]. Interestingly, no cultural SES were discussed with respect to tropical

classification of SES

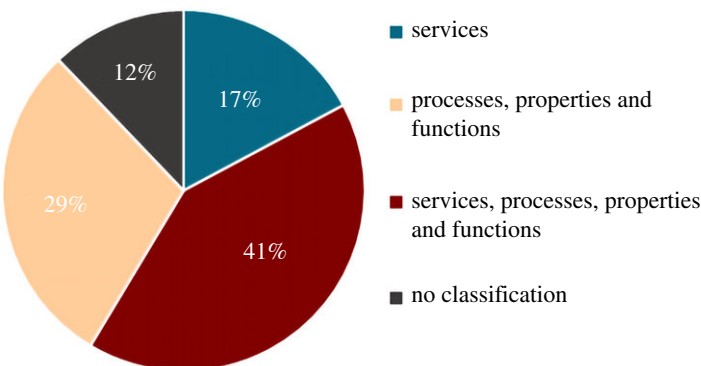

**Figure 3.** Classification of SES according to the articles in the database.

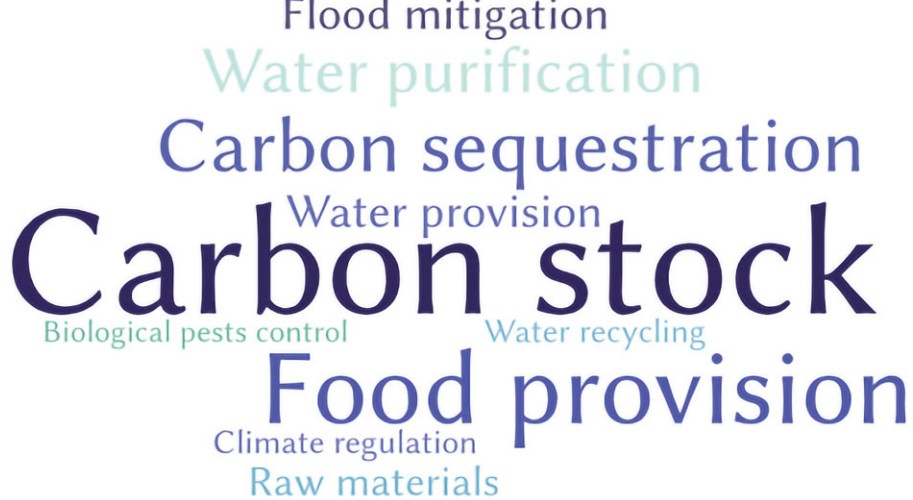

**Figure 4.** Most addressed SES found in the systematic review.

regions. This observation was surprising given the enormous diversity of cultures in tropical regions and the presence of anthropogenically modified soils (e.g. Amazonian Dark Earths and soil modified by charcoal kilns in the Brazilian Atlantic Forest) [38–41]. A low number of publications referring to cultural SES has been reported in previous publications [6] and might be explained by the fact that SES is often recognized as supporting other ES [4]. In the discussion of ES, cultural services have been the least addressed in the literature [42]. This observation highlights the need for a holistic SES approach that simultaneously considers the multiple services delivered by soil and the cultural value of soil.

## 3.2. Assessing and valuating soil ecosystem services

The assessment of SES in tropical regions was proposed by eight studies out of the 41 analysed (table 1). These studies showed different interpretations of the ES concept and the natural capital of soil. Some studies ($n = 3$) assessed soil C stock ES and undertook soil natural capital stock measurements (example: C stored in collected soil samples; table 1). Soil diagnoses based only on measurements of the soil's natural capital present one face of the soil condition and not the service operation. Other studies reported models that simulated SES based upon both static (e.g. information on soil natural capital stocks, properties and processes) and dynamic variables (e.g. how properties/processes vary temporally) [9,51]. Only one study in our database considered dynamic variables (time scale); in this instance, to measure flood mitigation and C stock [48]. It is highlighted that out with the studies included in the database, several publications focused upon the temperate region, considered SES assessment using models (static and dynamic variables), for example: modelling soil ecosystem services (MOSES) [52], soil carbon, aggregation, structure turnover (CAST) (used for the SoilTrEC

**Table 1.** Studies, SES evaluated according to the authors and description of the methods.

| reference | soil ES assessment | methods characterization |
|---|---|---|
| Ditt EH et al. [43] | water filtration | modelling |
| Lathuilliere MJ et al. [44] | water purification and climate regulation | modelling |
| Maass JM et al. [45] | food provision | interview |
| Marichal R et al. [46] | C stock | one field visit/laboratory analysis |
| Trilleras JM et al. [47] | C stock and food provision | one field visit/laboratory analysis |
| Campos A et al. [48] | C stock and flood mitigation | several field visits/laboratory analysis/modelling |
| Chanlabut U et al. [49] | C stock | one field visit/laboratory analysis |
| Wood SLR et al. [50] | food and timber | one field visit/interview |

project [53,54]) and AgriPolis [55,56]. Given this combined approach, we believe it is essential that future studies that seek to assess SES in tropical regions consider both static and dynamic variables.

An attempt to undertake the valuation of SES in a tropical region was noted in three of the 41 studies [57–59]. In these studies, services such as water purification [57], rubber plantation [58] and soil processes such as erosion control and nutrient cycling were valued [59]. Studies seeking to value carbon stock and carbon sequestration have been developed but have not yet been applied in tropical regions. Methods that estimate the economic value of climate regulation are based on various factors, such as cost of carbon sequestration in various contexts based on the market price of carbon or the willingness to pay for increased soil carbon sequestration [60]. For soil carbon storage valuation, Keith et al. [61] propose several options, including 'land value', 'avoided carbon stock loss' and 'value of a fixed asset on the balance sheet' [61]. Many articles on SES published in recent years address the idea of assigning prices to soil services, but rarely do they propose a number, nor do they suggest methods that can be used to perform the valuation [19,62].

In this area of quantitative SES (e)valuation, much more effort is required. We recommend a transition from the assessment of soil's natural capital and process to a framework that pulls these elements through to consider the multiple services delivered by soil. Evaluation of SES can contribute to a holistic view of land use, a framework to appraise synergies and trade-offs and a lens through which to focus on solutions and policies to optimize the supply of SES.

## 3.3. Temporal and geographical distribution of case studies

The number of publications on SES in the tropical region has increased in recent years, especially over the last 6 years (figure 5). This observation follows the global trend, in which the multidisciplinary scientific interest in the ES theme has grown, due to the attention given by the governments and international agencies to soil conservation as a necessity for human well-being [3,63]. A review of the soil quality concept indicated that from 1970 to 2010, the concept was related to soil productivity (mainly for agriculture) [64]. Currently, the concept of soil quality is related to the provision of ES and soil multifunctionality [64].

Of all the countries located in the tropical region, only 15 studies connected soils to ES (figure 6). This relatively low number may be related to the coupled-biases linked to only a small amount of funding for research directed towards SES in the tropical region; this limiting the acquisition of data and restricting the size of the database with which to support the evaluation of SES [65]. In most tropical countries, there are studies that classified SES inconsistently, i.e. they also classified the SES as properties, processes and functions (figure 6).

Of the total of 41 studies evaluated, the tropical region of the South American continent showed the largest number of case studies (Brazil = 8, Colombia = 6 and Mexico = 4) (figure 6). This higher value in Brazil ($n = 8$) may be related to its role as one of the world's largest agricultural producers [66] and also to the growing recognition of the ES concept in Brazil. Furthermore, environmental services promoted by thematic research networks developed in recent years [67], with emphasis on the Sustainable Amazon Network [68], Brazilian Agricultural Research Corporation—Embrapa's Environmental Services Portfolio [69], Biota Fapesp [70] and the Brazilian Platform for Biodiversity and Ecosystem Services (BPBES) [71] may have underpinned the more significant number of cases.

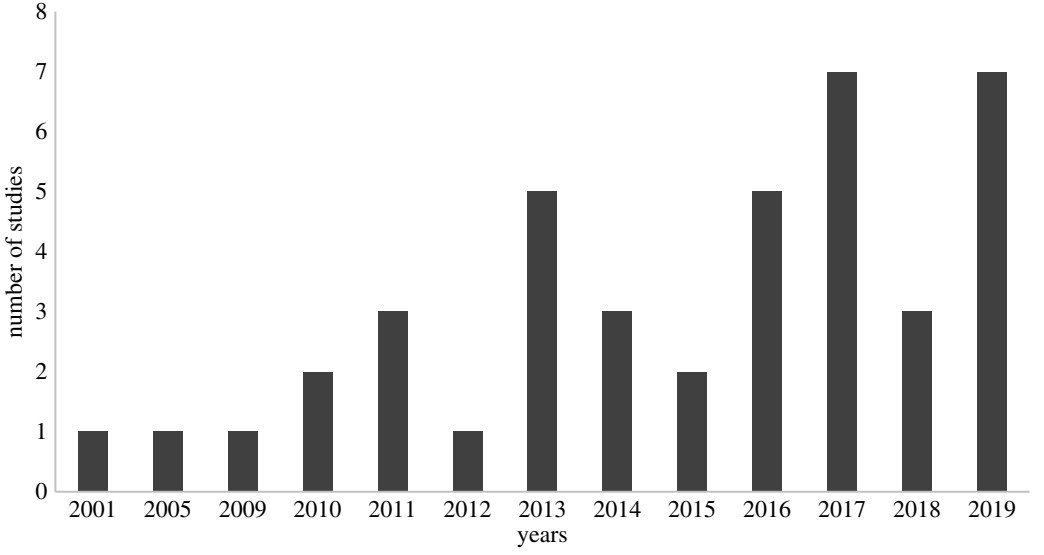

**Figure 5.** Number of studies relating SES over time in the tropical region.

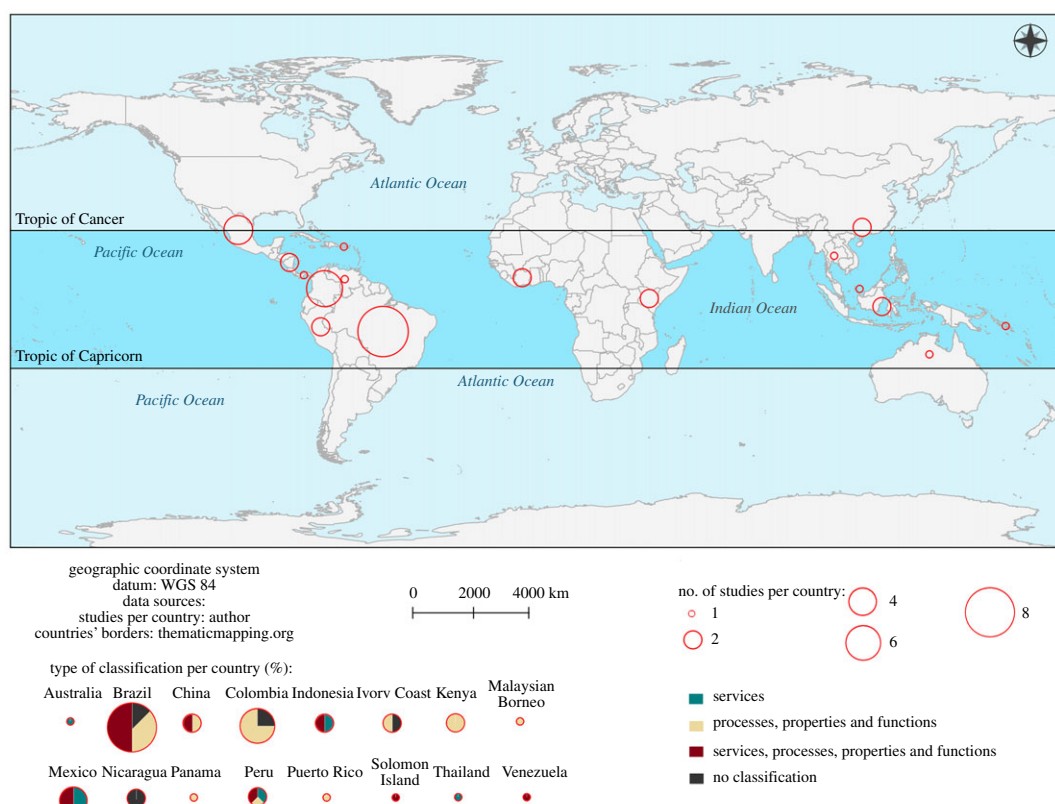

**Figure 6.** Map summarizing the case studies linked to countries in the tropical region. The colours in the pie charts depict how SES were defined.

The tropical region of the African continent showed two case studies (Kenya = 2) and the Asian continent four case studies (Indonesia = 2, Thailand = 1 and Malaysia = 1). Kenya has been gaining prominence in the African continent on the theme of ES [72]. The low number of publications in the tropical region of Africa and Asia highlight that more investment in research directed towards the SES theme is needed.

The SES theme can help in the development of policy instruments such as payments for SES [10,73,74]. These mechanisms can encourage ecosystems conservation and restoration, initiatives that are crucial in the countries of the tropical region. More knowledge regarding SES would help these

countries overcome the severe degradation of the natural capital of soil they suffer and improve their socio-economic growth [75,76].

## 4. Conclusion

To our knowledge, this is the first systematic review regarding SES in the tropical region. Although the number of publications devoted to SES and the tropical region has increased from 2001 to 2019 (from 1 to 41), the number is still small. This finding, in itself, highlights a persisting knowledge gap and underscores that further research to capture SES data for the tropical region and its interpretation is needed.

The temporal distribution of case studies indicates that the number of publications on SES in the tropical region has increased in recent years, especially over the last 6 years. In terms of geography, tropical South America stood out, and Brazil particularly so, with the greatest number of aligned studies. Of the 41 articles evaluated, 36 provided some classification of SES, and the majority (41%) classified SES as soil services, processes, properties and functions. Only eight studies assessed and valuated SES. They showed different interpretations of the ES concept and the natural capital of soil. We propose that future research dedicated to SES in a tropical context should clearly define and adequately apply the concept of ES to the soil.

We assert that soil diagnoses based only on measurements of the soil's natural capital present one face of the soil condition; but, this is an incomplete view. We hope that funding agencies support more SES projects so that classification and methods will become more robust. It is also necessary to develop suitable soils databases of the tropical region for the use of the scientific community dedicated to working with the SES. We recommend a transition from the assessment of soil properties and soil function to a framework that pulls these elements through to consider the multiple services delivered by soil, including the cultural values. Evaluation of SES has the potential to contribute to a holistic view of land use, a framework to appraise synergies and trade-offs and a lens through which to focus on solutions and policies to optimize the supply of SES. However, efforts are required to develop policy instruments, such as payments for SES, that will incentivize the conservation and restoration of ecosystems, maintaining and enhancing the associated SES. We therefore urge scientists, farmers and governments in tropical regions to collaborate and to work towards a common goal of holistic appreciation of SES.

Data accessibility. The datasets supporting this article have been uploaded as part of the electronic supplementary material.

Authors' contributions. A.F.R. and A.E.L. conceived of and designed the study. A.F.R. wrote the first draft of the manuscript. A.F.R., A.E.L., B.J.R., A.E.S., J.L.V.-V., K.A.K., M.C., N.F.F. and V.D. developed and discussed the Material and methods section. A.F.R. did the data extraction. A.F.R., A.E.L., B.J.R., A.E.S., C.L., E.C.C.F., F.R.S., F.T., I.P., J.L.V.-V., K.A.K., M.C., N.F.F., R.B.P., V.M., V.D. and W.G.T. helped with subsequent drafts. All authors gave final approval for publication.

Competing interests. We have no competing interests.

Funding. This study was supported by Royal Society/ Newton Advanced Fellowship grant no. NAF\R2\180676, Coordenação de Aperfeiçoamento de Pessoal – Brasil (CAPES) Finance Code 001, Carlos Chagas Foundation for Research Support of the State of Rio de Janeiro (FAPERJ) no. E-26/202.680/2018, National Council for Scientific and Technological Development (CNPq) no. 308536/2018-5 and Pontifical Catholic University of Rio de Janeiro – Scholarship to encourage productivity in teaching and research 2018–2020.

Acknowledgements. We thank the Royal Society/ Newton Fund for the grant which enabled the organization of the two workshops 'Back to the roots: value that grows from the land' that allowed the authors to meet each other, and *Coordenação de Aperfeiçoamento de Pessoal de Nível Superior—Brasil* (CAPES) for the stipend of the first author and the graduate course. Additionally, we thank the Carlos Chagas Foundation for Research Support of the State of Rio de Janeiro (FAPERJ), the National Council for Scientific and Technological Development (CNPq) and Pontifical Catholic University of Rio de Janeiro—Scholarship to encourage productivity in teaching and research 2018–2020. Eric Lino by the map. We also thank the anonymous reviewers for their constructive comments and suggestions.

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
