## [Peer Review File · Royal Society Open Science]

Review History

RSOS-201584.R0 (Original submission)

Review form: Reviewer 1

Is the manuscript scientifically sound in its present form?

No

Are the interpretations and conclusions justified by the results?

Yes

Is the language acceptable?

Yes

Do you have any ethical concerns with this paper?

No

Reports © 2021 The Reviewers; Decision Letters © 2021 The Reviewers and Editors; Responses © 2021 The Reviewers, Editors and Authors. Published by the Royal Society under the terms of the Creative Commons Attribution License <http://creativecommons.org/licenses/by/4.0/>, which permits unrestricted use, provided the original author and source are credited

Have you any concerns about statistical analyses in this paper?

Yes

Recommendation?

Major revision is needed (please make suggestions in comments)

Comments to the Author(s)

The manuscript "Systematic review of soil ecosystem services in tropical region" explores the scientific production about soil ecosystem services in the tropical region. Despite the manuscript is very relevant to the field, I consider that some modifications would improve the text. The manuscript is short, thus maybe by exploring the reasons behind choices would become the text clearer.

My main concern is that there is a disconnection between the search performed and the structure of the manuscript. I understand that the authors would like to explore the importance of the science-policy interface in SES discussion, but this topic appears isolated in the introduction/methods. How the outcome of the search contribute to the next steps in SES studies? What are the mechanisms that could explain the differences observed (methodological issues)? Are there some coupled-biases related to SES and countries? If yes, why? Do studies in the tropical region present a different profile of temperate region? In which context SES are explored in the manuscripts? Impacts of land-use changes? Deforestation? Restoration? These are some of the several questions that authors could explore better and would improve the manuscript. In its current form, the manuscript sounds more a report about the number of papers in each country.

In my opinion, the introduction would be better if organized as follow:

1st paragraph: SES and their importance (General).

2nd paragraph: SES and the conceptual background behind them (Figure 1). I would also add some information/discussion about the use of the nature contribution to people concept.

3rd paragraph: Factors and human-induced changes that can affect the ability of soil to provide services (Risks related).

4th paragraph: How those changes are more pronounced in tropical regions and how it can compromise the ES provided by the region (Demand and urgency). GAP: There is no systematic review that presents how SES is being explored in the tropical region.

5th paragraph: Goal, methodological strategy (briefly). Potential of these results to decision-makers.

In the methods section, it is not clear why authors have used Portuguese as an alternative language. This is confusing because there are other very important languages in the tropical region that are not described in the methods. If the inclusion of Portuguese did not improve the number of manuscripts used in the review, I would remove these search terms from the methods. Did authors think to use country names as an alternative search term to the tropical region? Most authors do not emphasize this territorial aspect.

Did the authors think to use some statistical analysis to reinforce their arguments? Some generalized linear mixed models are a good strategy.

If authors decide to keep this simple (only exploring country and ES type information), I recommend improving the discussion that explains the patterns explored.

The temporal behaviour of SES is different from other ES? Is there something special in SES if we compare with other ecosystems?

I liked the figures but I imagined a mixture of Figure 3 and Figure 4: a single figure by substituting the red circles in Figure 3 by coloured circles as described in Figure 4.

I am not convinced that only 8 studies assessed the SES in the tropical region (Section 4.3). Do these results include modelling papers using software such as INVEST? I would expect a greater number of returns. Maybe the number is related to the search terms used (tropical region?). Did

the authors think to use a diagram to demonstrate the flow of choices made in their systematic review?

I like the conclusion but I would reinforce the arguments previously in the discussion section. Supplementary Material: I would organize better the .xls file to be attached as Supplementary Material. Authors do not need to show all the processed information. Please, be sure that all information is in English and consistently presented. A single table seems to be enough to summarize all information collected.

Review form: Reviewer 2

Is the manuscript scientifically sound in its present form?

Yes

Are the interpretations and conclusions justified by the results?

Yes

Is the language acceptable?

Yes

Do you have any ethical concerns with this paper?

No

Have you any concerns about statistical analyses in this paper?

No

Recommendation?

Accept with minor revision (please list in comments)

Comments to the Author(s)

The paper does a literature review and characterization of published studies evaluating aspects of soils in the context of ecosystem services in the tropics. The results show an increase over time in relevant published studies, the geographic distribution of studies and they quantify the relative magnitude of different soil functions that are identified/focused on in the papers. The main conclusion is that there is presently a lack of a consistent framework to define soil ecosystem services, which hamper communication between scientists/practitioners and policy makers.

The paper is clearly written and easy to follow. The main point about inconsistencies in classification systems/characterizations is clear. However, I feel the paper is lacking in providing more concrete proposals about how to improve SES assessments in the future to better inform policy makers. A few sentences in the Conclusions allude to this, e.g., "We recommend a transition from the assessment of soil properties and soil function to a framework that pulls these elements through to consider the services delivered by soil" - what does that mean?? I think the authors could have some valuable insights to provide in terms of a way forward, but they stop short of working to better articulate such a vision.

Additional specific points:

Title: Suggest changing to something like "Systematic review of soil ecosystem service classifications in tropical regions (they've review classification systems, not the services per se)

In Figure 1, it should probably be ("N₂O and CH₄")

Decision letter (RSOS-201584.R0)

Dear Dr Rodrigues

The Editors assigned to your paper RSOS-201584 "Systematic review of soil ecosystem services in tropical regions" have now received comments from reviewers and would like you to revise the paper in accordance with the reviewer comments and any comments from the Editors. Please note this decision does not guarantee eventual acceptance.

Please submit your revised manuscript and required files (see below) no later than 21 days from today's (ie 10-Dec-2020) date. Note: the ScholarOne system will 'lock' if submission of the revision is attempted 21 or more days after the deadline. If you do not think you will be able to meet this deadline please contact the editorial office immediately.

on behalf of Professor Pete Smith (Subject Editor)
openscience@royalsociety.org

Reviewer comments to Author:

Reviewer: 1

Comments to the Author(s)

The manuscript "Systematic review of soil ecosystem services in tropical

region” explores the scientific production about soil ecosystem services in the tropical region. Despite the manuscript is very relevant to the field, I consider that some modifications would improve the text. The manuscript is short, thus maybe by exploring the reasons behind choices would become the text clearer.

My main concern is that there is a disconnection between the search performed and the structure of the manuscript. I understand that the authors would like to explore the importance of the science-policy interface in SES discussion, but this topic appears isolated in the introduction/methods. How the outcome of the search contribute to the next steps in SES studies? What are the mechanisms that could explain the differences observed (methodological issues)? Are there some coupled-biases related to SES and countries? If yes, why? Do studies in the tropical region present a different profile of temperate region? In which context SES are explored in the manuscripts? Impacts of land-use changes? Deforestation? Restoration? These are some of the several questions that authors could explore better and would improve the manuscript. In its current form, the manuscript sounds more a report about the number of papers in each country.

In my opinion, the introduction would be better if organized as follow:

1st paragraph: SES and their importance (General).

2nd paragraph: SES and the conceptual background behind them (Figure 1). I would also add some information/discussion about the use of the nature contribution to people concept.

3rd paragraph: Factors and human-induced changes that can affect the ability of soil to provide services (Risks related).

4th paragraph: How those changes are more pronounced in tropical regions and how it can compromise the ES provided by the region (Demand and urgency). GAP: There is no systematic review that presents how SES is being explored in the tropical region.

5th paragraph: Goal, methodological strategy (briefly). Potential of these results to decision-makers.

In the methods section, it is not clear why authors have used Portuguese as an alternative language. This is confusing because there are other very important languages in the tropical region that are not described in the methods. If the inclusion of Portuguese did not improve the number of manuscripts used in the review, I would remove these search terms from the methods. Did authors think to use country names as an alternative search term to the tropical region? Most authors do not emphasize this territorial aspect.

Did the authors think to use some statistical analysis to reinforce their arguments? Some generalized linear mixed models are a good strategy.

If authors decide to keep this simple (only exploring country and ES type information), I recommend improving the discussion that explains the patterns explored.

The temporal behaviour of SES is different from other ES? Is there something special in SES if we compare with other ecosystems?

I liked the figures but I imagined a mixture of Figure 3 and Figure 4: a single figure by substituting the red circles in Figure 3 by coloured circles as described in Figure 4.

I am not convinced that only 8 studies assessed the SES in the tropical region (Section 4.3). Do these results include modelling papers using software such as INVEST? I would expect a greater number of returns. Maybe the number is related to the search terms used (tropical region?). Did the authors think to use a diagram to demonstrate the flow of choices made in their systematic review?

I like the conclusion but I would reinforce the arguments previously in the discussion section. Supplementary Material: I would organize better the .xls file to be attached as Supplementary Material. Authors do not need to show all the processed information. Please, be sure that all information is in English and consistently presented. A single table seems to be enough to summarize all information collected.

Reviewer: 2
 Comments to the Author(s)

The paper does a literature review and characterization of published studies evaluating aspects of soils in the context of ecosystem services in the tropics. The results show an increase over time in relevant published studies, the geographic distribution of studies and they quantify the relative magnitude of different soil functions that are identified/focused on in the papers. The main conclusion is that there is presently a lack of a consistent framework to define soil ecosystem services, which hamper communication between scientists/practitioners and policy makers.

The paper is clearly written and easy to follow. The main point about inconsistencies in classification systems/characterizations is clear. However, I feel the paper is lacking in providing more concrete proposals about how to improve SES assessments in the future to better inform policy makers. A few sentences in the Conclusions allude to this, e.g., "We recommend a transition from the assessment of soil properties and soil function to a framework that pulls these elements through to consider the services delivered by soil" - what does that mean?? I think the authors could have some valuable insights to provide in terms of a way forward, but they stop short of working to better articulate such a vision.

Additional specific points:

Title: Suggest changing to something like "Systematic review of soil ecosystem service classifications in tropical regions (they've review classification systems, not the services per se)

In Figure 1, it should probably be ("N₂O and CH₄")

===PREPARING YOUR MANUSCRIPT===

If you have been asked to revise the written English in your submission as a condition of publication, you must do so, and you are expected to provide evidence that you have received language editing support. The journal would prefer that you use a professional language editing service and provide a certificate of editing, but a signed letter from a colleague who is a native speaker of English is acceptable. Note the journal has arranged a number of discounts for authors

using professional language editing services
(<https://royalsociety.org/journals/authors/benefits/language-editing/>).

===PREPARING YOUR REVISION IN SCHOLARONE===

<https://royalsociety.org/journals/authors/author-guidelines/#supplementary-material> to include a suitable title and informative caption. An example of appropriate titling and captioning may be found at https://figshare.com/articles/Table_S2_from_Is_there_a_trade-

off_between_peak_performance_and_performance_breadth_across_temperatures_for_aerobic_sc
ope_in_teleost_fishes_/3843624.

Author's Response to Decision Letter for (RSOS-201584.R0)

See Appendix A.

RSOS-201584.R1 (Revision)

Review form: Reviewer 1

Is the manuscript scientifically sound in its present form?

Yes

Are the interpretations and conclusions justified by the results?

Yes

Is the language acceptable?

Yes

Do you have any ethical concerns with this paper?

No

Have you any concerns about statistical analyses in this paper?

Yes

Recommendation?

Accept as is

Comments to the Author(s)

The manuscript "Systematic review of soil ecosystem services in tropical regions" is much improved in its current form. I do appreciate the work performed by the authors in answering the comments previously made in the last review. The figure 6 is richer than its previous version, despite I would also represent circle size in the legend (type of classification by country) and I would remove country names from map. Please, be sure that at the final version figures will be at the appropriate resolution and double-check for typos in the text (i.e. "eighth" P7,L48).

I have to say that an statistical analysis would turn the argumentation more sophisticated and stronger, but I understand the choice to keep the analysis as they are. In fact, there are several systematic reviews that perform a more qualitative discussion, based only on porcentages and general patterns.

I consider it is a great contribution to tropical SES scientists and it will be relevant to RSOS readers.

Decision letter (RSOS-201584.R1)

Dear Dr Rodrigues

On behalf of the Editors, we are pleased to inform you that your Manuscript RSOS-201584.R1 "Systematic review of soil ecosystem services in tropical regions" has been accepted for publication in Royal Society Open Science subject to minor revision in accordance with the referees' reports. Please find the referees' comments along with any feedback from the Editors below my signature.

Please submit your revised manuscript and required files (see below) no later than 7 days from today's (ie 23-Feb-2021) date. Note: the ScholarOne system will 'lock' if submission of the revision is attempted 7 or more days after the deadline. If you do not think you will be able to meet this deadline please contact the editorial office immediately.

on behalf of Prof Pete Smith (Subject Editor)
openscience@royalsociety.org

Reviewer comments to Author:
Reviewer: 1

Comments to the Author(s)
The manuscript "Systematic review of soil ecosystem services in tropical

regions" is much improved in its current form. I do appreciate the work performed by the authors in answering the comments previously made in the last review. The figure 6 is richer than its previous version, despite I would also represent circle size in the legend (type of classification by country) and I would remove country names from map. Please, be sure that at the final version figures will be at the appropriate resolution and double-check for typos in the text (i.e. "eighth" P7,L48).

I have to say that an statistical analysis would turn the argumentation more sophisticated and stronger, but I understand the choice to keep the analysis as they are. In fact, there are several systematic reviews that perform a more qualitative discussion, based only on porcentages and general patterns.

I consider it is a great contribution to tropical SES scientists and it will be relevant to RSOS readers.

===PREPARING YOUR MANUSCRIPT===

===PREPARING YOUR REVISION IN SCHOLARONE===

-- Ensure that your data access statement meets the requirements at <https://royalsociety.org/journals/authors/author-guidelines/#data>. You should ensure that you cite the dataset in your reference list. If you have deposited data etc in the Dryad repository, please only include the 'For publication' link at this stage. You should remove the 'For review' link.

Author's Response to Decision Letter for (RSOS-201584.R1)

See Appendix B.

Decision letter (RSOS-201584.R2)

Dear Dr Rodrigues,

It is a pleasure to accept your manuscript entitled "Systematic review of soil ecosystem services in tropical regions" in its current form for publication in Royal Society Open Science.

Appendix A

Review - Systematic review of soil ecosystem services in tropical regions

Reviewer 1

Comments	Reply	Action in the manuscript
1- My main concern is that there is a disconnection between the search performed and the structure of the manuscript. I understand that the authors would like to explore the importance of the science-policy interface in SES discussion, but this topic appears isolated in the introduction/methods. How the outcome of the search contribute to the next steps in SES studies?	We thank the reviewer for the comment and we acknowledge that the abstract and the introduction may have generated an expectation to discuss the aspect of better dialogue between science and decision-making, which is not the manuscript focus. Scientific studies rarely address SES in the tropical region. Our study seeks to contribute by identifying the gaps and problems related to SES research in this region. This research effort also aims to encourage future studies in the tropical region on SES, promoting more robust critical view on evaluating and valuing soil services.	Sentences inserted in the introduction: “Through a systematic literature review, we assessed the state of the art of SES in tropical regions. In order to guide future research and to encourage more robust critical view on evaluating and valuing soil services this study identifies knowledge gaps and problems associated with the use of the SES concept in the tropical region”.
2- What are the mechanisms that could explain the differences observed (methodological issues)?	We recognize that some differences that we have identified are related to the methodological issues of studies with SES. Of all the countries located in the tropical region, only 15 presented studies that link soils to ecosystem services. This low number may be related to the low number of funding for research aimed at the SES, which limit the acquisition of data in the field, for example; the low number and, sometimes, the absence of a database with adequate data for the evaluation of the SES. Concerning the most addressed SES in the studies, the differences can be explained for more significant market interest in the supply of food, inventory and carbon sequestration, when compared to other services.	Sentences inserted in the results and discussion, section 4.3: “Of all the countries located in the tropical region, only 15 studies connected soils to ES (figure 6). This relatively low number may be related to the coupled-biases linked to only a small amount of funding for research directed towards SES in the tropical region; this limiting the acquisition of data and restricting the size of the database which to support the evaluation of SES (59). In most tropical countries there are studies that classified SES inconsistently, i.e. they also classified the SES as properties, processes and functions (figure 6).” Sentences inserted in the results and discussion, section 4.1: “Also, the market interest in the supply of food, carbon stock and sequestration may have contributed to greater interest in these services.”

3- Are there some coupled-biases related to SES and countries? If yes, why?	Yes, there may be some coupled-biases related to countries and SES. They actually may be linked to methodological issues or conceptualization of the SES (connected to the previous reviewer comment). We added an additional sentence in the manuscript highlighting this information for our readers.	Sentences inserted in the results and discussion, section 4.3: “This relatively low number may be related to the coupled-biases linked to only a small amount of funding for research directed towards SES in the tropical region; this limiting the acquisition of data and restricting the size of the database which to support the evaluation of SES (59).”
4- Do studies in the tropical region present a different profile of temperate region?	We are not sure what the reviewer means by ‘profile’. If we understand correctly the question, the response is yes and this is one of the reasons we decided to carry out this review with a focus on the tropical region. Prior to this review, we found out that the majority of the studies on SES are produced and carried out in the temperate region (some references to support this claim: Dominati et al. 2014; Adhikari et al. 2016; Baveye et al. 2016; Jonsson et al. 2016; Brady et a. 2019; among others; complete reference at the end of this document). In the tropical region, there are still few studies on the concept’s discussion (Prado et al. 2015) and few evaluation (Campos et al. 2011) and valuation initiatives (Portela et al. 2001). We believe this rationale is already explained in the manuscript.	
5- In which context SES are explored in the manuscripts? Impacts of land-use changes? Deforestation? Restoration? These are some of the several questions that authors could explore better and would improve the manuscript.	We thank the reviewer to bringing this point up. The article seeks to explore SES in the tropical region in all contexts mentioned by the reviewer. We added more information to the manuscript.	Sentences inserted in the introduction: “The high demand for services provided by soils has transformed the global landscape (1). Natural ecosystems have been degraded and/or replaced by agricultural production, mining activities and cities (25). The impacts of such changes are projected to worsen with increasing global population, which, according to United Nations projections, may reach 9.7 billion inhabitants in 2050 (26). Sustainable soil management along with conservation and restoration initiatives, alongside changing society’s consumption habits, are urgently needed to assure the provision of SES. The absence of effective measures to prevent further soil degradation will lead to intensify climate change, hunger and the spread of diseases (3,5,6).” Sentences inserted in the conclusion:

		“...efforts are required to develop policy instruments, such as payments for SES, that will incentivise the conservation and restoration of ecosystems, maintaining and enhancing the associated SES.”
6- In its current form, the manuscript sounds more a report about the number of papers in each country. In my opinion, the introduction would be better if organized as follow: 1st paragraph: SES and their importance (General). 2nd paragraph: SES and the conceptual background behind them (Figure 1). I would also add some information/discussion about the use of the nature contribution to people concept. 3rd paragraph: Factors and human-induced changes that can affect the ability of soil to provide services (Risks related). 4th paragraph: How those changes are more pronounced in tropical regions and how it can compromise the ES provided by the region (Demand and urgency). GAP: There is no systematic review that presents how SES is being explored in the tropical region. 5th paragraph: Goal, methodological strategy (briefly). Potential of these results to decision-makers.	The authors are grateful for the reviewer's suggestion. We added the new introduction structure in the manuscript.	We changed the structure of introduction accordingly.
7- In the methods section, it is not clear why authors have used Portuguese as an alternative language. This is confusing because there are other very important languages in the tropical region that are not described in the methods. If the inclusion of Portuguese did not improve the number of manuscripts used in the review, I would remove these search terms from the methods.	We are grateful for the reviewer's suggestion and we will remove the Portuguese as none of the databases found articles when the search was made in Portuguese. See also the response to the following comment.	

8- Did authors think to use country names as an alternative search term to the tropical region? Most authors do not emphasize this territorial aspect.	We did indeed consider and tested various key words including using alternative names of the countries in the search. However, there are more than 90 countries in the tropical region. At some point we needed to decide about the scope of the study and that decision included the selection of our key words linked to the objectives of this study as presented in the methodology section. We chose to use the terms “tropical region”, because the expression soils in the tropical region is widely used in the literature. We acknowledge that this is a limitation of the study, as the most of the review studies have, and we now highlight this limitation in our manuscript.	Sentence inserted in the methodology section: “We recognize that the term ‘tropical region’ may have limited the search, as many articles use other spatial cut-outs (example country names) to define their locations. However, there a more than 90 countries in the tropical region and considering all of them as search terms would render this systematic review unfeasible from a practical standpoint.”
9- Did the authors think to use some statistical analysis to reinforce their arguments? Some generalized linear mixed models are a good strategy. If authors decide to keep this simple (only exploring country and ES type information), I recommend improving the discussion that explains the patterns explored.	We thank the reviewers for these observations, but we prefer to keep the methodology in the original way. We believe that it reflects the objective of this study well and leads to robust analysis and conclusions. Our approach is similar to other published review studies, for example, Gurwick et al. 2013; Mendes et al. 2018; and Ramirez-Agudelo et al. 2020.	
10- The temporal behaviour of SES is different from other ES? Is there something special in SES if we compare with other ecosystems?	Yes, it is. From different perspectives: 1) Soil formation is a very slow process (up to 1cm/100 years approx.) (Alexandrovsky, 2007). Therefore, when removing soil, or destroying its physical structure, the natural soil processes would take decades to build-up again the regenerate the soil layer. However, when applying sustainable management practices in agriculture (e.g. conservation agriculture, no-tillage...) they can foster these soil formation processes in already degraded soils or prevent soil degradation. 2) The “corner stone” of soil functioning (Smith et al. 2019) and, therefore, soil ecosystem services delivering, is the SOC content. SOC content is directly related to the improvement in the physical properties (e.g. decreasing soil bulk density, while increasing porosity, water holding capacity...), microbiological (e.g. increase in microbial activity) and chemical (e.g. linked to the increase in soil N and other soil nutrients). However, SOC content has a saturation limit, depending on the soil texture (i.e. amount of silt and clay) (Six	

	et al. 2002) and after some time – about 30 years in agricultural soils – under the same management or conditions it reaches the steady state (i.e. equilibrium) (Stewart et al. 2007) and, therefore, the amount of SOC that can be accumulated is limited. Thus, in agricultural soils, when shifting to sustainable management practices, the SOC sequestration rate is typically much higher during the first years after changing the management, whereas it decreases over time (Vicente-Vicente et al. 2016). Therefore, although the natural process of soil formation is very slow and can take up millennia to create 1cm of soil, soil formation processes can be enhanced through applying sustainable management practices in agriculture and forestry.	
11- I liked the figures but I imagined a mixture of Figure 3 and Figure 4: a single figure by substituting the red circles in Figure 3 by coloured circles as described in Figure 4.	We appreciate the reviewer's suggestion and we inserted the new map. We also chose to keep the graph that shows how the evaluated studies group classifies the SES. Then, we have two different results, general and per country. We chose to insert in the graph a category called “no classification” to refer to the five articles that related the concept of SES to aspects of the soil that do not fit into any of the categories services, properties, processes and functions of the soil.	We changed the figure accordingly.
12- I am not convinced that only 8 studies assessed the SES in the tropical region (Section 4.3). Do these results include modelling papers using software such as INVEST? I would expect a greater number of returns. Maybe the number is related to the search terms used (tropical region?).	Yes, these studies include work with the INVEST tool (see supplementary table material, articles id 44, 68 and 91). We redid the search using only the keywords “ecosystem services AND soil AND INVEST” considering the publications until April 2019, and we found a total of 79 articles. We checked the articles and only three would meet all the criteria, which does not alter the general view of the studies that have been developed on SES in the tropical region - for example, none of them made monetary valuation. Thus, we chose to highlight the limitation of the search term “tropical region” in the manuscript in the methodology.	

13- Did the authors think to use a diagram to demonstrate the flow of choices made in their systematic review?	We are thankful for the reviewer's suggestion.	A diagram demonstrating the flow of choice in the systematic review was inserted in the methodology section.
14- I like the conclusion but I would reinforce the arguments previously in the discussion section.	We thank the reviewer for this comment and we edited accordingly.	Sentences inserted in the result and discussion section 4.2: “Soil diagnoses based only on measurements of the soil's natural capital present one face of the soil condition and not the service operation”. “In this area of quantitative SES (e)valuation, much more effort is required. We recommend a transition from the assessment of soil’s natural capital and process to a framework that pulls these elements through to consider the multiple services delivered by soil. Evaluation of SES can contribute to a holistic view of land use, a framework to appraise synergies and trade-offs and a lens through which to focus on solutions and policies to optimise the supply of SES.” Sentences inserted in the result and discussion section 4.3: “SES theme can help in the development of policy instruments such as payments for SES (67,68). These mechanisms can encourage ecosystems conservation and restoration, initiatives that are crucial in the countries of the tropical region. More knowledge regarding SES would help these countries overcome the severe degradation of the natural capital of soil they suffer and improve their socio-economic growth (69,70). “
15- Supplementary Material: I would organize better the .xls file to be attached as Supplementary Material. Authors do not need to show all the processed information. Please, be sure that all information is in English and consistently presented. A single table seems to be enough to summarize all information collected.	We thank the reviewer for this comment and for the thorough revision of the manuscript and the supplementary materials.	Corrected accordingly

Reviewer 2

16- The main conclusion is that there is presently a lack of a consistent framework to define soil ecosystem services, which hamper communication between scientists/practitioners and policy makers. The paper is clearly written and easy to follow. The main point about inconsistencies in classification systems/characterizations is clear. However, I feel the paper is lacking in providing more concrete proposals about how to improve SES assessments in the future to better inform policy makers. A few sentences in the Conclusions allude to this, e.g., “We recommend a transition from the assessment of soil properties and soil function to a framework that pulls these elements through to consider the services delivered by soil” – what does that mean?? I think the authors could have some valuable insights to provide in terms of a way forward, but they stop short of working to better articulate such a vision.	We thank the reviewer for the comment and added into the manuscript alternatives to improve the SES assessments in the future. We suggest that funding agencies should support more projects aimed at discussing and developing SES assessment methods and schemes because the more tested in practice, the more robust the methods become; and the development suitable soils databases of the tropical region for the use of scientific community dedicated to working with SES. We believe that evaluating soil properties and functions in the context of SES, has the potential to promote more significant changes in land management. The SES approach was proposed to highlight the vital role of well-managed soil for human well-being, translating scientific results in a relevant format do decision-making and to instigate a better dialogue between science and decision-makers.	Sentences inserted in the conclusion: “We hope that funding agencies support more SES projects so that classification and methods will become more robust. It is also necessary to develop suitable soils databases of the tropical region for the use of the scientific community dedicated to working with the SES.”
17- Additional specific points: Title: Suggest changing to something like “Systematic review of soil ecosystem service classifications in tropical regions (they’ve review classification systems, not the services per se)	We thank the reviewer for suggesting a new title. However, we believe that the article does not only address how studies have classified SES, but presents a broader view of what is being published in the tropical region. The article presents the number of publications per year, the location of where more studies are being produced in the tropical region, and the range of initiatives related to evaluation and monetary valuation of SES.	
18- In Figure 1, it should probably be (“N2O and CH4”)	We are thankful for the reviewer's comment and for the time dedicated to revise our manuscript.	We changed accordingly.

References

- Adhikari K, Hartemink AE. Geoderma Linking soils to ecosystem services — A global review. *Geoderma*. 2016;262:101–11. Available from: <http://dx.doi.org/10.1016/j.geoderma.2015.08.009>
- Adolfo Campos C, Hernández ME, Moreno-Casasola P, Espinosa EC, Alejandra Robledo R, Mata DI. Soil water retention and carbon pools in tropical forested wetlands and marshes of the Gulf of Mexico. *Hydrol Sci J*. 2011;56(8):1388–406. Available from: <https://doi.org/10.1080/02626667.2011.629786>
- Alexandrovskiy, Alexander L. Rates of soil-forming processes in three main models of pedogenesis. *Rev. mex. cienc. Geol*. 2007, 24, (2): 283-292.
- De Almeida GCA. Serviços Ecológicos do solo sob sistemas agroflorestais: estado da arte e estudo de caso em São Gonçalo. UNIVERSIDADE FEDERAL FLUMINENSE; 2019. Available from: <https://www.alice.cnptia.embrapa.br/alice/bitstream/doc/1109924/1/DissertacaoGustavoCesarAraujoDeAlmeida2019.pdf>
- Baveye PC, Baveye J, Gowdy J. Soil “ecosystem” services and natural capital: Critical appraisal of research on uncertain ground. *Front Environ Sci*. 2016;4 (1)–49. Available from: <https://doi.org/10.3389/fenvs.2016.00041>.
- Brady M V., Hristov J, Wilhelmsson F, Hedlund K. Roadmap for valuing soil ecosystem services to inform multi-level decision-making in agriculture. *Sustain*. 2019;11(19):1–20. Available from: [doi:10.3390/su1119528](https://doi.org/10.3390/su1119528)
- Dominati E, Mackay A, Green S, Patterson M. A soil change-based methodology for the quantification and valuation of ecosystem services from agro-ecosystems: A case study of pastoral agriculture in New Zealand. *Ecol Econ*. 2014;100:119–29. Available from: <http://dx.doi.org/10.1016/j.ecolecon.2014.02.008>
- Gurwick NP, Moore LA, Kelly C, Elias P. A Systematic Review of Biochar Research, with a Focus on Its Stability *in situ* and Its Promise as a Climate Mitigation Strategy. *PLoS ONE*. 2013; 8:9. Available from: <https://doi.org/10.1371/journal.pone.0075932>
- Jónsson JÖG, Davíðsdóttir B, Nikolaidis NP. Valuation of Soil Ecosystem Services. *Advances in Agronomy*. 2017; 142:353-384. Available from: <http://dx.doi.org/10.1016/bs.agron.2016.10.011>
- Mendes M, Latawiec, AE, Sansevero JBB, Crouzeilles R, Moraes LF, Castro A, Alves-Pinto HN, Brancalion PHS, Rodrigues RR, Chazdon RL, Barros FSM, Santos J, Iribarrem A, Mata S, Lemgruber L, Rodrigues A, Korys K, Strassburg BBN. Look down – there is a gap – the need to include soil data in Atlantic Forest restoration. *Restoration Ecology*. 2018; 27: 2, 361-370. Available from: [doi:10.1111/rec.12875](https://doi.org/10.1111/rec.12875)
- Portela R, Rademacher I. A dynamic model of patterns of deforestation and their effect on the ability of the Brazilian Amazonia to provide ecosystem services. *Ecol Modell*. 2001;143(1–2):115–46. Available from: [https://doi.org/10.1016/S0304-3800\(01\)00359-3](https://doi.org/10.1016/S0304-3800(01)00359-3)
- Ramírez-Agudelo, N.A.; Porcar Anento, R.; Villares, M.; Roca, E. Nature-Based Solutions for Water Management in Peri-Urban Areas: Barriers and Lessons Learned from Implementation Experiences. *Sustainability* 2020; 12, 9799. Available from: <https://doi.org/10.3390/su12239799>
- Six, J, Conant, RT, Paul, EA *et al*. Stabilization mechanisms of soil organic matter: Implications for C-saturation of soils. *Plant and Soil*. 2002; 241, 155–176, Available from: <https://doi.org/10.1023/A:1016125726789>
- Smith P, Adams J, Beerling DJ, Beringer T, Calvin KV, Fuss S, Griscom B, Hagemann N, Kammann C, Kraxner F, Minx JC, Popp A, Renforth P, Vicente-Vicente JL, Keesstra S. Land-Management Options for Greenhouse Gas Removal and Their Impacts on Ecosystem Services and the Sustainable Development Goals. *Annual Review of Environment and Resources*. 2019; 44:255-286. Available from: <https://doi.org/10.1146/annurev-environ-101718-033129>
- Stewart CE, Paustian K, Conant RT. *et al*. Soil carbon saturation: concept, evidence and evaluation. *Biogeochemistry*. 2007; 86, 19–31. Available from: <https://doi.org/10.1007/s10533-007-9140-0>
- Vicente-Vicente JL, García-Ruiz R, Francaviglia R, Aguilera E, Smith P. Soil carbon sequestration rates under Mediterranean woody crops using recommended management practices: A meta-analysis. *Agriculture, Ecosystems & Environment*. 2016; 235, 204-214. Available from: <https://doi.org/10.1016/j.agee.2016.10.024>.

Appendix B

Review - Systematic review of soil ecosystem services in tropical regions

Reviewer 1

Comments	Reply	Action in the manuscript
1- The manuscript "Systematic review of soil ecosystem services in tropical regions" is much improved in its current form. I do appreciate the work performed by the authors in answering the comments previously made in the last review. The figure 6 is richer than its previous version, despite I would also represent circle size in the legend (type of classification by country) and I would remove country names from map. Please, be sure that at the final version figures will be at the appropriate resolution and double-check for typos in the text (i.e. "eighth" P7,L48). I have to say that an statistical analysis would turn the argumentation more sophisticated and stronger, but I understand the choice to keep the analysis as they are. In fact, there are several systematic reviews that perform a more qualitative discussion, based only on percentages and general patterns. I consider it is a great contribution to tropical SES scientists and it will be relevant to RSOS readers.	We thank the reviewer for comments about the manuscript and suggestion for the figure 6.	We changed the figure accordingly.